# Peer review of "Refining Adjuvant Therapy for Endometrial Cancer: New Standards and Perspectives"

_biology, 2021, doi:10.3390/biology10090845_

Round 1

Reviewer 1 Report

 This review provides a comprehensive overview of endometrial carcinoma adjuvant therapy, yet some improvements are necessary.

Line 266, In the sentence “Despite the recent implementation of molecular in to clinical classification…”, a noun either “features” or “characteristics” should be added following “molecular”.

Lines 303-304, the sentence “RT is optional in both arm. The primary…” does not have to start a new paragraph, while it can be place in line 302 following the previous sentence.

Line 307, “a with more individualized targeted…” is awkward.

Author Response

This review provides a comprehensive overview of endometrial carcinoma adjuvant therapy, yet some improvements are necessary.

Line 266, In the sentence “Despite the recent implementation of molecular in to clinical classification…”, a noun either “features” or “characteristics” should be added following “molecular”.

Authors Response 1: Dear Reviewer, thanks for your suggestion to improve our manuscript. The sentence has been modified accordingly (see line 256, page 7 of the marked manuscript)

Lines 303-304, the sentence “RT is optional in both arm. The primary…” does not have to start a new paragraph, while it can be place in line 302 following the previous sentence.

Authors Response 2: Thanks for your observation. The sentence has been changed as you suggest (see lines 288-291, page 9 of the marked manuscript)

Line 307, “a with more individualized targeted…” is awkward.

Authors Response 3: Thanks for this consideration. The sentence has been modified (see lines 293-294, page 9 of the marked manuscript)

Reviewer 2 Report

The manuscript has been revised satisfactorily.

Two minor suggestion.

Line 125 the word "prize" can be replaced by "cost".

Line 206...it has to be noticed can be replaced by ... It has to be noted or - it should be noted.

Author Response

Reviewer's comments:

The manuscript has been revised satisfactorily.

Two minor suggestion.

1) Line 125 the word "prize" can be replaced by "cost".

Authors Response 1: Dear Reviewer, thanks for your suggestion to improve our manuscript. The sentence has been changed as you suggest (see line 120, page 4 of the marked manuscript).

Line 206...it has to be noticed can be replaced by ... It has to be noted or - it should be noted.

Authors Response 2: Thanks for your observation. The sentence has been changed as you suggest (see lines 197, page 5 of the marked manuscript)

Reviewer 3 Report

It is a well-written review discussing the approach to endometrial cancer treatment based on the new ESGO/ESTRO/ESP guidelines together with some limitations of guidelines which demand further RCT studies. It better suits to the book than Biology journal, however, substantive level of the manuscript can not be neglected. I have no remarks to the text, especially after improvements made by the authors..

Author Response

Reviewer's comments:

It is a well-written review discussing the approach to endometrial cancer treatment based on the new ESGO/ESTRO/ESP guidelines together with some limitations of guidelines which demand further RCT studies. It better suits to the book than Biology journal, however, substantive level of the manuscript can not be neglected. I have no remarks to the text, especially after improvements made by the authors..

Authors Response 1: Dear Reviewer, thanks for your nice comments.

This manuscript is a resubmission of an earlier submission. The following is a list of the peer review reports and author responses from that submission.

Round 1

Reviewer 1 Report

An important trial targeting HER2 over-expressed endometrial cancer (EC) should be added. A long-term follow-up of prospective phase II trial revealed that trastuzumab in combination with carboplatin and paclitaxel prolonged progression-free survival (PFS: HR 0.44, 95% CI 0.23-0.83, p = 0.015) and overall survival (OS: HR 0.49, 95% CI 0.25-0.97, p = 0.041) than chemotherapy alone especially in stage III, IV HER2 over-expressed serous EC (Clin Cancer Res 2020; 26:3928-3935). 

Reviewer 2 Report

Histology/grade, Myometrial invasion, LVSI and lymph-node involvement are well established prognostic factors for endometrial cancer. Molecular markers are increasingly recognized as (better) prognostic markers. It should be recognized that these although newer molecular markers appears to be qualitatively associated with prognosis have never been tasted for there independent prognostic value in relation to LVSI and nodes! Surely a serous carcinoma without LVSI and nodes even in presence of NSMP/MMRd serous marker will have a better prognosis than LVSI and node positive tumour without NSMP/MMRd serous. It appears to me that the poor prognostic molecular markers have relative prognostic significance and not an independent significance that can be tested in a multivariate model. Generally speaking a patient who has positive nodes usually recommended post-operative radiotherapy. In the same vain, will a patient with NSMP/MMRd serous will always require chemotherapy, even in absence of LVSI and nodes? Surely the relapse rates in serous with LVSI and node negative, LVSI positive and node positive patients are going to be vastly different. I think the circumstances under which the molecular markers may determine adjuvant treatment should be simplified, explained and where possible supported by evidence.

Reviewer 3 Report

This is a non-systematic review by a research group that is not leading in this field. The review does not add to the recently published ESGO/ESTRO/ESP guideline. So it is a bit difficult to determine the added value of publishing this manuscript. It's strongest points are: 1) the compactness, it is more attractive to read than the guideline itself and 2) the discussion of running and future trials that test adjuvant treatment strategies according to molecular class.

So I suggest to make this paper more interesting and citable to work out those 2 strengths more by:

  • add a table of graph that is very comprehensible and shows the clinician what treatment should be give to which group of patients. Make this paper the quick and handy summary of the treatment of EC.
  • write a separate section or add a table with all little subgroups summed up for which the 2021 guideline does not provide recommendations due to lack of evidence (e.g. stage IA p53 abn, malignant polyps etc..)
  • Improve and extend the perspectives section. Be complete on running and upcoming trials (do a systematic search in the trial registries). Point out to the reader what questions will be answered by these trials and identify gaps and open question that could be addressed in future studies.

Besides this, there are many large and small points that need to be edited. See below.

Introduction

  • please use more recent figures for the incidence of EC, e.g. globocan stats for 2020 are available.
  • Aging is of the population an important factor for the increasing incidence of EC, please add.
  • EC in 15-44 yr old women is extremely rare, I don't think mentioning their 5 yr OS contributes to this review as EC in youngsters is not discussed elsewhere
  • line 43 onwards: You may discuss outdated systems for classification of EC to give historical perspective, but current phrasing is not correct. Do not state that there are 3 ways to classify EC, do not suggest that classifying EC in type 1 and 2 is still an acceptable method.
  • Clarify in line 49-50 that the 5th WHO classification is the current one (2020) and that it includes both histopatholgical features and molecular class.
  • Line 57-58. The TCGA analyses were very important indeed, but it had no immediate therapeutic application. The TCGA classification requires to perform copy-number analysis, which is not feasible in clinical practice. Surrogate markers have been developed and validated to get to an assessable EC classification. Beyond that, prospective clincial trials are required to determine whether adjuvant treatment can be changed according to molecular class.
  • line 59: The description of the 4 molecular classes is not in line with current knowledge "copy number low (endometroid), 59 and copy number high (serous- like)" Several studies have shown that histotypes such as endometrioid and serous are not confined to certain molecular classes. There are NSMP EC non-endometrioid histology and p53 mutant ECs with endometriooid histology.
  • The section on the surrogate markers for the 4 TCGA classes should be moved upwards, before the statement that molecular testing is acceptable. Add the appropriate references of the Canadian and Dutch groups who have developed and validated the surrogate markers.
  • The section on multiple classifiers is not up to date, the studies that the others refer to have been adopted and included in the latest guidelines: it is known which 11 POLE mutations are pathogenic, and that EC with such mutations are always classified as POLEmut regardless of p53 and MMR status. MMRd EC with secondary p53 are classified as MMRd. 

  • Line 116: the PORTEC-3 analysis of POLE should that prognosis was excellent independent of the addition of chemotherapy to EBRT, that is not the same as independent of treatment. The prognosis of POLE patients without any adjuvant treatment cannot be deducted from the PORTEC 3 study.
  • Line 143-144: [26] This is not a recent study at all, it's from 2013, and it still classified ECs as type 2. Please rephrase
  • Line 168-174: please start the discussion of the Maggi trial with it's main outcome and then any subanalysis on risk factors.
  • The section that discusses the literature on HIR EC refers to several studies that have included both HIR and HR patients. The authors should make clearer which of the outcomes are applicable to the current definition of HIR and where the uncertainties are.
  • Line 229-231: this statement is not coherent. PORTEC3 did not show significant benefit (and only a very minor difference of 2%) of adding chemotherapy to EBRT in stage I–II endometrial cancer. In other words, the addition of CT is probably not worth it. This does not imply that we should prescribe CT only! this implies that EBRT only is better than EBRT+CT. We can't deduct the value of CT only vs EBRT only from PORTEC-3 as this was not investigated.
  • The authors seem to overlook one of the most prominent gaps in current knowledge on treatment of HR EC. GOG 258 compared CT vs EBRT+CT, so also with the results of this trial we can't determine the value of CT only vs EBRT only in early stage HR EC. There are different interpretations in the world on combined outcomes of PORTEC3 and GOG258. This review should make this clear.

Perspectives

  • Are the authors aware that the molecular classification is currently being investigated in the PORTEC4a trial? This must be added. More details on the trial can be found in the trial paper: van den Heerik ASVM, Horeweg N, Nout RA, et al. Int J Gynecol Cancer 2020;30:2002–2007.
  • The correct abbreviation is RAINBO (no W). Please correct the names of the 4 trials too: POLEmut-BLUE, MMRd-GREEN, p53abn-RED and the NSMP-ORANGE trials. The GREEN, RED and ORANGE trials are RCTs.
  • It is not certain yet that RED will be placebo controlled, so it is better remove this part of the phrase.
  • GREEN will randomize between EBRT (probably + placebo) and EBRT plus the PD-L1 (not PD-1) inhibitor durvalumab, during and for 1 year after EBRT. Only MMRd EC stage III and stage II with substantial LVSI are eligible.
  • ORANGE will not be placebo controlled as it compares the addition of CT to EBRT with the addition of HT to EBRT. 
  • Please add that the overarching goal of the RAINBO trial is to compare standard treatment to personalized treatment based on molecular features. For this, data of all 4 trials will be pooled. This aspect makes the RAINBO program more than 4 independent trials.
  • Line 259: The authors state that there are 3 trials investigating immunotherapy and subsequently discuss only 1. Please correct this.
  • Line 267: It is not clear to which trials this phrase refers to as it is placed in the section of the ENGOT-EN11 trial. 'It is expected that these trials, if positive, will completely change 267 the adjuvant treatment strategy of EC.' Beyond that, the wording is speculative, please rephrase.

Reviewer 4 Report

The review is well written, however, the information presented in it has been already implemented for clinical practice, so novelty of the paper is at least doubtful.